# Assessing the Clinical Impact of the SARS-CoV-2 Gamma Variant on Intensive Care Unit Admissions: Insights from a Reference Hospital in Northeastern Brazil

**DOI:** 10.3390/v16030467

**Published:** 2024-03-20

**Authors:** Carolina Kymie Vasques Nonaka, Adlas Michel de Jesus Ribeiro, Gisele Vieira Rocha, Helena Souza da Hora, Antônio Augusto Fonseca Junior, Fernanda de Macêdo Lima, Iasmin Nogueira Bastos, Samara Alves Sa Teles, Thamires Gomes Lopes Weber, Vanessa Ferreira Costa, Zaquer Suzana Costa-Ferro, Clarissa Araújo Gurgel Rocha, Silvia Inês Sardi, Gúbio Soares, Ana Verena Almeida Mendes, Bruno Solano de Freitas Souza

**Affiliations:** 1D’Or Institute for Research and Education (IDOR), Salvador 41253-190, BA, Brazil; carolina.nonaka@hsr.com.br (C.K.V.N.); givieiraroch@gmail.com (G.V.R.); zaquercosta@yahoo.com.br (Z.S.C.-F.); clarissa.gurgel@fiocruz.br (C.A.G.R.); 2São Rafael Hospital, Salvador 41253-190, BA, Brazil; samara.teles@hsr.com.br (S.A.S.T.); thamires.weber@hsr.com.br (T.G.L.W.); vanessa.fcosta@hsr.com.br (V.F.C.); ana.verena@rededor.com.br (A.V.A.M.); 3Institute of Health Sciences, Federal University of Bahia, Salvador 40231-300, BA, Brazil; adlas.barbosa1@gmail.com (A.M.d.J.R.); gubiosoares@gmail.com (G.S.); 4Gonçalo Moniz Institute, FIOCRUZ, Salvador 40296-710, BA, Brazil; 5National Agropecuary Laboratory, MAPA, Pedro Leopoldo 33250-220, MG, Brazil; antonio.biomol@gmail.com; 6Bahiana School of Medicine Salvador, Salvador 40290-000, BA, Brazil; fernandalima0799@gmail.com; 7Faculty of Medicine and Dentistry, Federal University of Bahia, Salvador 40110-100, BA, Brazil; inbastos3@gmail.com; 8Instituto Gonçalo Moniz, Fundação Oswaldo Cruz, Rua Waldemar Falcão, 121, Salvador 40296-710, BA, Brazil

**Keywords:** SARS-CoV-2, gamma, coronavirus disease-19 (COVID-19)

## Abstract

The global challenge posed by the prolonged COVID-19 pandemic underscores the critical need for ongoing genomic surveillance to identify emerging variants and formulate effective public health strategies. This retrospective observational study, conducted in a reference hospital in Northeast Brazil and comprising 2116 cases, employed PCR genotyping together with epidemiological data to elucidate the impact of the Gamma variant during its emergence, revealing distinct patterns in hospitalization rates, severity of illness, and outcomes. The study emphasizes the challenges posed by the variant, particularly an increased tendency for ICU admissions and respiratory support, especially among adults aged 18 to 59 without comorbidities. Laboratory analyses further demonstrate elevated inflammatory, coagulation, and hepatic markers in the Gamma variant cohort, suggesting a more severe systemic response. Despite limitations, including a retrospective approach and single-institution data, the study underscores the importance of ongoing genomic surveillance. Overall, this research contributes valuable insights into the impact of the Gamma variant on COVID-19 dynamics, advocating for continued research and surveillance to inform effective public health strategies regarding evolving viral variants.

## 1. Introduction

The COVID-19 pandemic has posed a significant global public health challenge, marking one of the most profound crises in the 21st century thus far. The establishment of an ongoing genomic surveillance initiative is crucial for promptly identifying emerging variants and formulating effective policies and strategies for mitigating the spread of SARS-CoV-2 and potential future infectious diseases. Since its emergence in Wuhan in December 2019, the SARS-CoV-2 virus has undergone considerable evolution. Despite genetic proofreading mechanisms that limit sequence diversity in coronaviruses, including SARS-CoV-2, natural selection can still act on occasional advantageous mutations. The COVID-19 pandemic contributed to the accumulation of immunologically significant mutations, potentially altering the virus’s transmission, pathogenesis, and antigenic properties. Notably, the gene encoding the spike protein emerged as a major focal point for modifications within the viral genome. Furthermore, it exhibited a high degree of conformational freedom, implying that the ability of the S component to adopt different shapes may play a role in the structural integrity or resilience of the virus [1,2,3].

The first reported mutation in the SARS-CoV-2, D614G in the spike protein, emerged in Germany and China in January 2020 [4]. The globally identified D614G mutation was associated with the emergence of new strains, along with notable mutations such as 501Y.V1 in England and deletions (ΔH69/ΔV70 and ΔY144), playing an instrumental role in classifying variants of concern (VOC) by the World Health Organization (WHO) [4,5,6,7]. These mutations pose a greater risk due to their potential for increased transmissibility, reinfection, severity, immune escape, and reduced vaccine efficacy [8,9]. Furthermore, mutations in the SARS-CoV-2 spike receptor binding domain (RDB) play a crucial role in direct interactions with the human ACE2 receptor, thereby facilitating host-cell invasion. Nonetheless, it is important to note that the lethality of most new viral strains has either remained constant or decreased, as indicated by prior research [4,7,10].

In genomic surveillance, accumulated mutations can serve as molecular markers for tracking geographically dispersed viruses. Examples include distinct lineages of SARS-CoV-2 with multiple spike protein mutations: Alpha in the UK, Beta in South Africa, and Gamma and P.2 in Brazil (both derived from the B.1.1.28 lineage) [6,11].

The Gamma variant, first identified in Amazonas, Brazil, gained international recognition during the second wave of the COVID-19 pandemic in January 2021 [12]. This variant displayed a gradual accumulation of mutations in the spike region, correlating with an increase in the number of cases of the disease [13]. Its emergence significantly impacted Brazil’s healthcare system, leading to a drastic rise in intensive care unit (ICU) bed occupancy and pushing several states to the brink of a healthcare collapse [14,15]. Notably, the Gamma variant was detected in Salvador, one of the largest cities in northeastern Brazil, around late December 2020 and early January 2021 [16]. This coincided with an alarming increase in hospital admissions and COVID-19-related deaths in the state of Bahia, particularly in February and March 2021. During this period, a notable demographic shift in ICU admissions was observed in a private hospital in Salvador [17]. Concurrently, other variants such as Alpha (B.1.1.7, lineage 20I) and Beta (B.1.351, lineage 20H) were also present in Brazil. However, according to data from the GISAID database, these variants had a lower prevalence compared to the Gamma variant, emphasizing the prevailing influence of Gamma in the region [18].

The capacity to track viral evolution plays a pivotal role in shaping public health strategies in response to the COVID-19 pandemic [19]. Monitoring the prevalence and dynamics of these VOCs carries significant clinical and epidemiological implications and is indispensable for constructing models that delineate the course of the pandemic and the patterns of mutation it undergoes. While Next Generation Sequencing (NGS) stands as the widely acknowledged gold standard for the genomic characterization of SARS-CoV-2 variants [20], its adoption remains encumbered by significant costs, particularly in countries like Brazil. The limited COVID-19 sequencing capabilities in low and middle-income countries have proven to be a critical challenge in the global fight against the pandemic, particularly with the emergence of VOCs. In early 2021, only a limited number of cases were confirmed through whole genomic sequencing for the identification of SARS-CoV-2 in Brazil [21]. Alternatively, PCR genotyping can be used for the detection of mutations as a more accessible and less expensive method to distinguish the Alpha variant from the Beta and Gamma variants [22,23,24]. 

Thus, this study employed PCR genotyping alongside epidemiological data to enhance the understanding of the epidemiological landscape, correlating it with clinical and laboratory data among COVID-19 patients from December 2020 to March 2021, during the presumed emergence of the SARS-CoV-2 Gamma lineage in Salvador, Northeast Brazil. It evaluated a cohort of 2116 individuals to profile those infected by the Gamma variant versus other circulating variants, including a notable increase in admissions to the intensive care unit (ICU) within a prominent private hospital located in Salvador, the fifth most populous capital city in Brazil.

## 2. Materials and Methods

Study design

A retrospective observational study was conducted at São Rafael Hospital, a private reference healthcare institution located in Salvador, Bahia, in the Northeast region of Brazil. Salvador ranks as the fifth most populous city in Brazil, with nearly 2.5 million inhabitants [25]. This study relied on data collected during the period from December 2020 to March 2021, coinciding with a notable escalation in hospital admissions related to COVID-19 and the introduction of the Gamma variant. The study included individuals attending and submitted to COVID-19 testing in the hospital unit, whether hospitalized or not, and encompassed all age groups.

Data collection

This study was reviewed by local IRBs and received ethical approval from the National Committee for Ethics on Research (CAAE: 46821621.5.0000.0048). The research dataset consisted of data on COVID-19 patients who underwent RT-PCR tests and received care at the São Rafael hospital, encompassing comprehensive details regarding their clinical, laboratory, and demographic profiles. This dataset, comprising 2116 cases, was obtained from the hospital’s health information system and electronic medical records. The inclusion and exclusion criteria are detailed in Appendix A. The level of severity was considered based on the need for hospitalization and respiratory support, following the criteria established by the World Health Organization [26]. Comorbidities were classified according to the criteria of the Brazilian Ministry of Health (high blood pressure, diabetes mellitus, lung diseases, cerebrovascular disease, chronic renal disease, immunosuppression, liver disease, and obesity). The laboratory parameters of 216 individuals admitted to the ICU were obtained from an electronic system used by the hospital. Additionally, data regarding confirmed COVID-19 cases in the city of Salvador and the state of Bahia were obtained from https://bi.saude.ba.gov.br (accessed on 9 December 2023).

RT-PCR Genotyping

The nasopharyngeal swab samples were initially processed by the laboratory’s standard protocols, using a validated commercial kit Allplex™ SARS-CoV-2 (Seegene, Seoul, Republic of Korea) following the methodology described previously by de Sousa et al., 2021 [27]. After the confirmation of SARS-CoV-2 detection, the samples, encoded by the laboratory system, were stored at −80 °C for subsequent assessment using RT-qPCR genotyping assay under conditions optimized by the study. The primers and probes utilized to discriminate the variants are described in Appendix A. The primers and probes were designed using Primer Express v2.0 software (Applied Biosystems, Foster City, CA, USA). This RT-qPCR genotyping was designed to detect two variants targeting the Δ3675–3677 SGF deletion in the ORF1a gene and the Δ69/70 HV deletion in the spike gene, thereby potentially identifying the Gamma variant (Brazil) and B.1.351 (South Africa). A concentration of 10 pmol/μL was standardized for the primers and 5 pmol/μL for the probe. Additionally, 5 μL of TaqPath™ 1-Step RT-qPCR master mix, CG (ThermoFisher Scientific, Waltham, MA, USA) was added, in a final volume of 15 μL. The amplification temperature protocol used the following steps: 25 °C for 2 min and 50 °C for 15 min for the cDNA synthesis step, 95 °C for 2 min, and 45 cycles at 95 °C for 3 s and 60 °C for 30 s in the ABI 7500 FAST thermocycler (Applied Biosystems). Alternatively, the TaqPath™ COVID-19 CE-IVD RT-PCR kit (TaqPath) (Thermofisher Scientific, Waltham, MA, USA) was employed according to the manufacturer’s recommendations. The N gene was used as an amplification standard for SARS-CoV-2 samples following the CDC 2019 Novel Coronavirus (2019-nCoV) [28].

Statistical analysis

The clinical characteristics, laboratory parameters, and outcomes were displayed in percentages and as median values along with interquartile ranges (IQR), representing central tendency and dispersion measures, respectively. The relationship between study groups was assessed utilizing the Pearson chi-square test and Wilcoxon rank-sum test, employing a 95% confidence interval (CI). Significance was established for *p*-values < 0.05. All statistical analyses were conducted using R (version 4.3.1).

## 3. Results

### 3.1. Clinical and Epidemiological Insights in a Northeast Brazilian Hospital during the Second Wave Linked to the Gamma Variant

Between December 2020 and March 2021, 2116 nasopharyngeal swab samples (out of a total of 21,618) underwent COVID-19 screening by RT-qPCR genotyping method, following criteria outlined in Appendix A. In Figure 1A, two significant peaks with an increase in confirmed cases of COVID-19 in the states of Bahia and the city of Salvador, commonly referred to as waves, are observed. A similar trend was also observed in a private reference hospital (São Rafael Hospital) situated in Salvador (Figure 1B). During the second wave, the SARS-CoV-2 Gamma lineage was identified in December 2020 and rapidly emerged as the dominant variant. This is illustrated in Figure 1C, representing samples sequenced within the study region and deposited in GISAID, along with the results from RT-qPCR genotyping conducted in this study.

Table 1 presents a comparative analysis of the clinical characteristics and outcomes between individuals infected with the Gamma variant (*n* = 754) and those with other COVID-19 variants (*n* = 1362), as identified through RT-qPCR genotyping assay. Both groups exhibited remarkably similar distributions in terms of gender and age. Approximately 80% of all confirmed COVID-19 cases did not require hospitalization. Similar clinical profiles and outcomes were observed for both groups. Patients infected with the Gamma variant presented a lower cycle threshold (Ct) value of the N gene, compared to the other variants (*p* < 0.01). A trend, though a non-statistically significant difference, indicated higher hospital admissions to both general wards (6.10% vs. 2.35%) and ICUs (11.94% vs. 9.25%) in individuals infected with the Gamma variant as compared to other variants. A particularly interesting trend was observed among adults aged 18 to 59 without comorbidities infected with the Gamma variant, who were more likely to require ICU care (37.78% vs. 27.78%). In contrast, patients with other variants exhibited a non-statistically significant trend toward a higher propensity for ICU admission, predominantly among elderly individuals (37.78% vs. 49.20%) and those with comorbidities at any age (62.22% vs. 72.22%). 

Among patients requiring intensive care, both groups—those infected with the Gamma variant and those with other variants—showed an increased BMI. However, there was no statistically significant difference in BMI between these two groups. The duration of ICU stay ranged from 2 to 15 days in both groups, and the mortality rates were similar, ranging from 10% to 11% within the group of patients requiring intensive care. Interestingly, a significant distinction was observed in patients who did not require respiratory support in the ICU (18.89% vs. 46.03%, *p* < 0.01), suggesting that those infected with the Gamma variant consistently required more intensive respiratory assistance compared to those with other variants. The death rate in the ICU did not show a significant difference between the Gamma variant and other variants. 

### 3.2. Assessment of ICU-Admitted COVID-19 Patients Stratified by the Gamma Variant and Other Variants in a Reference Hospital 

Table 2 presents a comparative analysis of laboratory parameters among ICU-hospitalized individuals, distinguishing between those with the Gamma variant (*n* = 90) and those with other variants (*n* = 126). The Gamma variant group showed distinct laboratory profiles with significant differences in several markers: inflammatory and coagulation markers (including D-dimer, activated partial thromboplastin time, and ferritin), tissue damage markers (Lactate Dehydrogenase), liver function markers (alkaline phosphatase and gamma-glutamyl transferase), muscle damage markers (creatine kinase), and renal markers (creatinine and urea). These disparities were evident when compared to patients infected with other COVID-19 variants. Additionally, the Gamma variant group exhibited slightly higher levels of C-reactive protein (CRP), with a *p*-value of 0.05. There was also a noticeable upward trend in cardiac injury markers, specifically NT-proBNP and Troponin, in the Gamma variant group as opposed to those with other variants.

In Table 3, we present a detailed comparative analysis focusing on COVID-19 patients admitted to the ICU, specifically comparing those infected with the Gamma variant to those with other variants. This analysis centers on the utilization of respiratory support as a key indicator of disease severity. Of these patients, 81% infected by the Gamma variant and 54% infected by other variants needed respiratory support. Despite the absence of significant differences in gender or age distribution, a higher proportion of male individuals were hospitalized compared to females requiring respiratory support. However, a trend suggests higher comorbidity rates among those with other variants who required respiratory support, while a larger proportion of individuals aged 18 to 59, and those with no comorbidities, infected with the Gamma variant, were present in the group necessitating respiratory support. Additionally, individuals infected with the Gamma variant exhibited longer symptom durations before hospital admission and shorter ICU stays. Interestingly, there were no significant variations in the use of mechanical ventilation between the groups requiring oxygen support. Significantly, patients with the Gamma variant requiring respiratory support displayed elevated levels of laboratory markers, indicating a more pronounced inflammatory, coagulation, hepatic, and renal response, as evidenced by elevated levels of C-reactive protein, D-dimer, activated partial thromboplastin time, lactate dehydrogenase, ferritin, alkaline phosphatase, creatine kinase, gamma-glutamyl transferase, alanine aminotransferase, aspartate aminotransferase, creatinine, and urea (*p* < 0.05).

## 4. Discussion

The emergence of new variants of SARS-CoV-2, such as the Gamma variant, has posed significant challenges to global health. This study aimed to explore the clinical and epidemiological implications of the Gamma variant in Northeast Brazil during the second wave of the COVID-19 pandemic in Brazil. The results provide valuable insight into the impact of the Gamma variant on hospitalization rates, severity of illness, and the need for intensive care. Furthermore, this study emphasizes the importance of genomic surveillance in tracking the prevalence of variants and understanding their implications for public health.

Previously, a multiplexed RT-qPCR method was published that was capable of differentiating the B.1.1.7, B.1.351 and P.1 variants of interest simultaneously [29]. The TaqPath COVID-19 assay from Applied Biosystems (ThermoFisher) was also utilized for screening the P.1 variant as a commercial option. In accordance with these protocols, this study utilized samples that were sequenced during the standardization of RT-PCR genotyping with 100% agreement. Furthermore, the distribution of variants in the study region was found to be consistent between the graphs of the sequenced samples and those reported by RT-PCR genotyping (Figure 1C,D). This study successfully employed RT-qPCR genotyping alongside epidemiological data to profile individuals infected with the Gamma variant, providing a comprehensive overview of the epidemiological landscape. The use of PCR genotyping, as a cost-effective alternative sequencing method, allowed for the efficient analysis of a large number of samples during a period when Brazil still faced challenges in conducting large-scale genomic monitoring [30]. The rise of the Gamma variant in samples from a reference private hospital, reflecting the trend in Salvador and across Brazil, coincided with an increase in hospitalizations, especially among adults. Additionally, it highlighted distinctions in the patient profile between the Gamma variant compared to other variants from December 2020 to March 2021. 

The clinical characteristics and outcomes of individuals infected with the Gamma variant were compared to those with other variants, revealing important distinctions. Notably, individuals infected with the Gamma variant showed a higher tendency for ICU admissions, especially among adults aged 18 to 59 without comorbidities. Moreover, the analysis of ICU patients needing respiratory support revealed that a greater percentage of adults without comorbidities infected with the Gamma variant required this support. Additionally, they exhibited elevated laboratory markers compared to those with other variants, suggesting a more pronounced inflammatory and respiratory response linked to the Gamma variant. In a parallel study conducted in southern Brazil with a more limited cohort, increased rates of advanced ventilatory support and mortality were observed among those with confirmed Gamma infections compared to non-Gamma-infected patients, implying a more severe clinical course [31]. This finding further suggests that non-elderly patients hospitalized for COVID-19 with the Gamma variant during the second wave exhibited greater disease severity. While a study in Brazil reported an increased risk for pediatric and adolescent individuals with COVID-19 caused by the Gamma variant [32], our study did not observe relevant information in this specific population.

Severe COVID-19 cases often trigger unchecked inflammation and cytokine storms. Assessing serum levels of pro-inflammatory cytokines can serve various purposes in managing COVID-19: risk evaluation, disease monitoring, prognosis, therapy selection, and treatment response prediction [33]. The study evaluated the laboratory parameters of ICU-admitted patients, providing a detailed comparison between those infected with the Gamma variant and those infected with other variants. The Gamma variant cohort exhibited distinct laboratory profiles, including elevated levels of inflammatory and coagulation markers. Moreover, some markers of organ damage were more elevated, suggesting a potential increase in the severity profile of the patients infected with the Gamma variant. These findings suggest a more severe and systemic response to infection with the Gamma variant, potentially contributing to increased ICU admissions. Few studies have systematically explored clinical and laboratory datasets related to the prevalence of the Gamma variant in Brazil. Previously, differences between individuals with moderate and severe COVID-19 were investigated in Northeast Brazil during the second wave [34]. Another study prior to the second wave highlighted pronounced alterations in inflammatory markers and comorbidities among critically ill elderly COVID-19 patients [35]. Cytokine analysis provided valuable insights into the immune response during COVID-19; however, interpreting the data necessitates careful consideration of the variability in cytokine profiles among patients, influenced by factors such as age, comorbidities, disease severity, and immune status [36].

Furthermore, the observed 1-log reduction in cycle threshold values in samples from patients infected with the Gamma variant implies an association with increased severity. This observation was made previously in the context of a comparison of real-time RT-PCR cycle threshold values with clinical features and severity among hospitalized patients in the first and second waves of the COVID-19 pandemic in India [37]. Nevertheless, it is crucial to emphasize that the cycle threshold value alone cannot be deemed a dependable surrogate marker of infectivity. Moreover, infectivity is also impacted by additional factors, such as host immunity. As a limitation of the study during this period, which also coincided with the beginning of vaccination in Brazil, there were no data available regarding the vaccination status of these patients. Notably, vaccination in Brazil began with the elderly population, and during the study period adults and young individuals had not yet received the vaccination.

The emergence of the Gamma variant and its specific propensity for propagation in Brazil, while not extending to other countries, remains an unresolved inquiry [38]. Characterized by its elevated viral load, the Gamma variant demonstrated heightened transmissibility and virulence in contrast to other circulating variants during that timeframe, showcasing consistent clinical and epidemiological patterns across Brazil [17,39]. Interestingly, subsequent variants, like the Omicron variant, have exhibited lower rates of hospitalization. Three reasons accounting for this include: (i) the benefits of SARS-CoV-2 vaccination; (ii) unique mutations within the Omicron variant not identified in earlier strains (N764K, D796Y, N856K, Q954H, N969K, and L981F); and (iii) the Omicron alters spike cleavage efficiency and reduces transmembrane serine protease 2 (TMPRSS2)-mediated entry, leading to diminished cell fusion and syncytial formation compared to the Alpha and Delta variants [40].

The study acknowledges the limitations of its retrospective approach and the use of a single healthcare institution’s data. The use of RT-PCR genotyping did not allow the stratification of the different variants that were present in the non-Gamma group, and the precise identification of the different variants that were circulating in that period would rely on sequencing data. The results presented herein, however, underscore the significance of ongoing genomic surveillance, especially in regions with limited resources. 

## 5. Conclusions

In summary, this study provides insights into the clinical and epidemiological impact of the Gamma variant in Northeast Brazil. The emergence of this variant during the second wave was associated with increased transmission and notable shifts in hospitalization patterns. The distinctive clinical and laboratory characteristics of individuals infected with the Gamma variant, along with elevated rates of ICU admissions and respiratory support, emphasize the potential severity linked to this lineage. The implementation of cost-effective methods like PCR genotyping can enhance the capacity to track variants and guide public health interventions. The COVID-19 pandemic has highlighted the importance of ongoing research and surveillance efforts in comprehending the dynamics of emerging variants and guiding effective public health strategies.

## Figures and Tables

**Figure 1 viruses-16-00467-f001:**
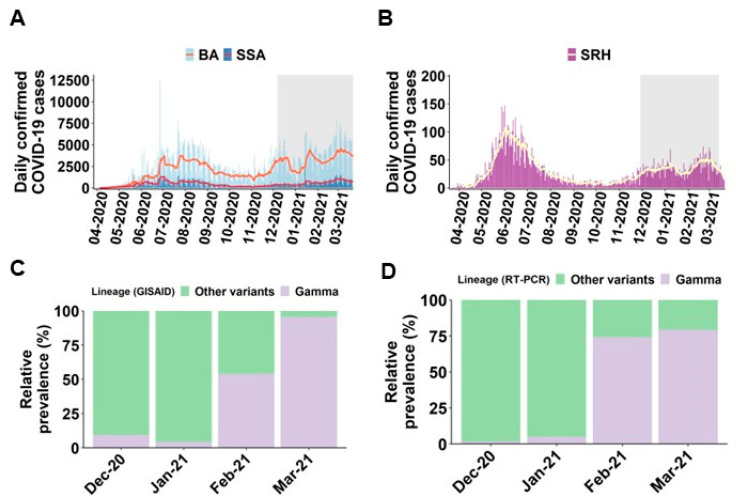
Distribution of Confirmed COVID-19 Cases by the Gamma Variant in Salvador, Bahia, During the Second Wave. (**A**) Daily confirmed COVID-19 cases in Bahia state (BA) and Salvador city (SSA) with a 7-day average available from Brazil surveillance agency. (**B**) Daily confirmed COVID-19 cases in São Rafael Hospital (SRH) with a 7-day average. (**C**) Relative frequency of Gamma compared to other variants in Salvador during the second wave using sequencing data (*n* = 263) available in GISAID and (**D**) the relative frequency of Gamma and other variants by RT-qPCR genotyping method in samples collected from patients who received care in São Rafael Hospital, a private hospital in Salvador (*n* = 2116).

**Table 1 viruses-16-00467-t001:** Clinical Characteristics and Outcomes of 2116 SARS-CoV-2 Infected Patients Between December 2020 to March 2021: Stratification by PCR Genotyping into Gamma Variant and Other Variants.

	Gamma *n* = 754	Other Variants *n* = 1362	*p*-Value
All patients			
Female *n* (%)	330 (43.77)	652 (47.87)	0.70
Male *n* (%)	419 (55.57)	699 (51.32)	0.62
did not inform sex *n* (%)	5 (0.66)	11 (0.81)	1.00
Age (years), median (IQR)	41 (33–54)	41 (32–54)	0.36
Median cycle thresholds (IQR)	16 (13–19)	17 (14–20)	**<0.01**
Non-Hospitalized			
Emergency/Laboratory	618 (81.96)	1204 (88.40)	0.60
Hospitalizations			
Ward			
Ward admissions *n* (%)	46 (6.10)	32 (2.35)	0.30
Age groups, ward *n* (%)			0.44
≥60	9 (19.60)	9 (28.10)	0.21
18–59	37 (80.40)	23 (71.90)	0.49
<18	0 (0.00)	0 (0.00)	-
Time on ward, median (IQR)	4.82 (3.44–6.66)	3.56 (2.47–5.73)	0.08
ICU			
ICU admissions *n* (%)	90 (11.94)	126 (9.25)	0.66
Time on ICU, median (IQR)	5.66 (1.92–15.00)	7.64 (2.04–15.6)	0.73
Age groups, UCI n (%)			0.15
≥60	34 (37.78)	62 (49.20)	0.25
18–59	55 (61.11)	61 (48.40)	0.24
<18	1 (1.11)	3 (2.38)	0.28
Comorbidities, ICU			0.15
yes *n* (%)	56 (62.22)	91 (72.22)	0.38
no *n* (%)	34 (37.78)	35 (27.78)	0.21
BMI, ICU median (IQR)	27.10 (24.10–31.00)	28.70 (25.60–31.60)	0.05
Non-invasive Respiratory support, ICU < 0.01			
supplementary oxygen > 3L, *n* (%)	30 (33.33)	30 (23.81)	0.19
supplementary oxygen <= 3L *n* (%)	43 (47.78)	38 (30.16)	0.05
none	17 (18.89)	58 (46.03)	**<0.01**
Mechanical ventilation, ICU			0.46
yes *n* (%)	31 (34.44)	37 (29.37)	0.47
no *n* (%)	59 (65.56)	89 (70.63)	0.66
Death, ICU			0.97
yes *n* (%)	9 (10.00)	14 (11.11)	0.81
no *n* (%)	81 (90.0)	112 (88.89)	0.93

Notes: Data are shown as median and interquartile (IQR) range or frequency (percentage). Categorical data were compared between the clinical groups using the Chi-squared tests. Continuous data were compared between the clinical groups using the Kruskal–Wallis test (for all groups). Bold font indicates statistical significance (*p* < 0.05). Abbreviations: BMI, Body Mass Index; ICU, intensive care unit.

**Table 2 viruses-16-00467-t002:** Laboratory Parameters Evaluated in Individuals with COVID-19 Infected with the Gamma or Other Variants Admitted to the ICU Between December 2020 and March 2021.

Parameters (Reference Value)	Gamma Median (IQR)	Other Variants Median (IQR)	*p*-Value
CRP (<10 mg/L)	47.80 (25.00–88.20)	42.55 (20.20–87.08)	0.05
D-dimer (<500 ng/mL)	1303 (817.00–2155.50)	1037 (567.00–1899.00)	**<0.01**
Fibrinogen (200 to 400 mg/dL)	550 (433.25–670.75)	543.50 (443.00–668.25)	0.64
APTT (25 to 40 seg)	20.00 (1.22–40.90)	11.99 (1.09–36.35)	**<0.01**
NT-proBNP (Age <50: 450 pg/mL, 50–75: 900 pg/mL, and >75: 1800 pg/mL)	217.50 (103.00–601.75)	213 (103.50–442.50)	0.07
Troponin 1, (<0.034 ng/mL)	0.07 (0.04–0.12)	0.03 (0.02–0.05)	0.06
LDH, (120 to 246 U/L)	347 (279.00–447.00)	287.50 (222.00–358.75)	**<0.01**
Ferritin, (17.9–464 ng/mL)	725.00 (398.00–1200.00)	547.30 (336.50–1022.50)	**<0.01**
Bilirubin, (0.2 to 1.3 mg/dL)	0.20 (0.00–0.40)	0.30 (0.00–0.40)	0.08
ALP, (38 to 126 U/L)	97 (74.00–146.25)	71 (54.00–102.75)	**<0.01**
CK, (Female: 30 to 135 U/L; Male: 55 to 170 U/L)	90 (45.00–219.50)	68 (40.50–143.00)	**<0.01**
GGT, (Female: 30 to 135 U/L; Male: 15 to 73 U/L)	175 (93.00–329.50)	121 (60.00–219.25)	**<0.01**
ALT, (Female: <35 U/L; Male: <50 U/L)	56 (33.00–91.00)	54 (34.00–84.00)	0.57
AST, (17 to 59 U/L)	46 (32–76)	40 (31–58)	**<0.01**
Creatinine, (0.7 to 1.2 mg/dL)	4.10 (3.05–4.95)	1.30 (1.00–1.40)	**<0.01**
Urea, (19 to 43 mg/dL)	47 (33.00–73.75)	44 (33.00–59.00)	**<0.01**

Notes: Data are shown as median and interquartile (IQR). Continuous data were compared between the clinical groups using the Wilcoxon Test (for all groups). Bold font indicates statistical significance (*p* < 0.05). Abbreviations: CPR, C-Reactive Protein; APTT, Activated Partial Thromboplastin Time; NT-proBNP, N-terminal prohormone of brain natriuretic peptide; LDH, Lactate Dehydrogenase; ALP, Alkaline Phosphatase; CK, Creatine Kinase; GGT, Gamma-glutamyl Transferase; ALT, Alanine Aminotransferase; AST, Aspartate aminotransferase.

**Table 3 viruses-16-00467-t003:** Comparative Analysis of Clinical Characteristics and Laboratory Parameters in Severe COVID-19 Cases Admitted to the ICU: Gamma Variant Versus Other Variants.

	Respiratory Support			Did Not Use		
	Gamma *n* = 73	Other *n* = 68	*p* Value	Gamma *n* = 17	Other *n* = 58	*p* Value
Sex, n (%)			0.94			0.73
Female	19 (26.00)	19 (27.94)	0.79	7 (41.20)	19 (32.80)	0.33
Male	54 (74.00)	49 (72.06)	0.87	10 (58.80)	39 (67.20)	0.45
Age, n (%)			0.23			0.46
≥60	29 (39.70)	34 (50.0)	0.28	5 (29.41)	28 (48.28)	0.83
18–59	44 (60.30)	34 (50.0)	0.33	11 (64.71)	27 (46.55)	0.08
>18	0 (0.00)	0 (0.00)	-	1 (5.88)	3 (5.17)	**0.03**
Comorbidity			0.16			0.43
yes, n (%)	47 (64.40)	52 (76.50)	0.31	9 (52.90)	39 (62.70)	0.19
no, n (%)	26 (35.60)	16 (23.50)	0.12	8 (47.10)	19 (32.80)	0.11
Days of symptoms	12 (8.00–19.00)	8 (5.00–11.50)	**<0.01**	4.5 (1.75–8.25)	7 (4.25–11.00)	**0.03**
Time on ICU, median (IQR)	7.49 (3.29–18.20)	12.30 (7.07–23.90)	**0.02**	1.00 (0.38–3.48)	2.12 (0.26–6.73)	0.27
MV, ICU			0.21			1.00
yes, n(%)	31 (42.5)	37 (54.40)		0 (0.00)	0 (0.00)	
no, n(%)	42 (57.5)	31 (45.60)		17 (100)	58 (100)	
Laboratory markers with reference value						
	median (IQR)			median (IQR)		
CRP (<10 mg/L)	48.20 (25.10–89.70)	42.60 (21.70–71.50)	**0.01**	39.95 (19.90–83.90)	47.10 (22.25–134.20)	0.28
D-dimer (<500 ng/mL)	1306 (824.00–2176.75)	1185 (673.50–1634.50)	**<0.01**	1186 (587.75–2011.75)	829 (546.50–1588.00)	0.08
Fibrinogen (200 to 400 mg/dL)	552 (433.25–672.00)	524 (439.25–614.50)	0.35	537 (433.00–664.25)	549 (463.75–676.75)	0.41
APTT (25 to 40 seg)	20.00 (1.22–41.00)	11.34 (1.10–36.92)	**<0.01**	11.99 (1.08–36.00)	12.89 (1.13–37.45)	0.77
NT-proBNP (Age <50: 450 pg/mL, 50–75: 900 pg/mL, and >75: 1800 pg/mL)	216.00 (102.00–620.00)	222.00 (147.50–367.00)	0.29	217.00 (107.50–503.00)	209.50 (94.5–338.75)	0.30
LDH (120 to 246 U/L	348.00 (281.00–448.00)	287.50 (204.25–415.50)	**<0.01**	277.00 (217.00–347.00)	311.00 (250.00–394.00)	0.31
Ferritin (17.9 to 464 ng/mL)	743.00 (409.25–1214.00)	543.00 (132.00–922.00)	**<0.01**	547.00 (354.00–1100.00)	545.00 (286.35–921.5)	0.53
Bilirubin (0.2 to 1.3 mg/dL)	0.2 (0–0.4)	0.2 (0–0.3)	0.06	0.3 (0–0.4)	0.2 (0–0.4)	**0.02**
ALP (38 to 126 U/L)	98.00 (74.00–148.00)	74.50 (58.50–91.50)	**<0.01**	71.00 (52.75–100.00)	71.00 (56.00–103.75)	0.77
CK (Female: 30 to 135 U/L; Male: 55 to 170 U/L)	91.00 (48.00–237.75)	51.00 (30.5–110.00)	**<0.01**	70.00 (40.50–134.50)	62.50 (41.50–195.75)	0.05
GGT (Female: 12 to 43 U/L; Male: 15 to 73 U/L)	180 (95–333.75)	84 (65–153.75)	**<0.01**	131 (64–242.00)	87 (44–172.00)	0.55
ALT (Female: <35 U/L; Male: <50 U/L)	55 (33.00–92.00)	58 (32.25–86.50)	0.31	57 (37.00–90.75)	48 (29.00–73.00)	0.19
AST (17 to 59 U/L)	46.00 (32.00–76.00)	37.00 (27.00–75.50)	**<0.01**	40.00 (30.00–57.00)	44.50 (33.00–58.00)	0.63
Creatinine (0.7 to 1.2 mg/dL)	4.10 (3.42–4.97)	0.70 (0.70–0.70)	**0.04**	1.09 (0.99–1.20)	1.30 (1.05–2.40)	0.25
Urea (19 to 43 mg/dL)	49 (35.00–76.00)	28 (21.50–38.00)	**<0.01**	45 (21.50–60.00)	40 (30.00–55.00)	**<0.01**

Notes: Data are shown as median and interquartile (IQR) range or frequency (percentage). Categorical data were compared between the clinical groups using the Chi-squared tests. Continuous data were compared between the clinical groups using the Kruskal–Wallis test (for all groups). Bold font indicates statistical significance (*p* < 0.05). Abbreviations: BMI, Body Mass Index; HBP, high blood pressure; ICU, intensive care unit; CPR, C-Reactive Protein; APTT, Activated Partial Thromboplastin Time; NT-proBNP, N-terminal prohormone of brain natriuretic peptide; LDH, Lactate Dehydrogenase; ALP, Alkaline Phosphatase; CK, Creatine Kinase; GGT, Gamma-glutamyl Transferase; ALT, Alanine Aminotransferase; AST, Aspartate aminotransferase.

## Data Availability

The clinical and laboratory data presented in this study originate from the confidential and restricted-access records of São Rafael Hospital.

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
