# Peer review of "Assessing the Clinical Impact of the SARS-CoV-2 Gamma Variant on Intensive Care Unit Admissions: Insights from a Reference Hospital in Northeastern Brazil"

_viruses, 2024, doi:10.3390/v16030467_

Round 1

Reviewer 1 Report

Comments and Suggestions for Authors

The authors conducted a retrospective observational study comprising 2,116 COVID-19 cases, employed PCR genotyping together with epidemiological data to elucidate the impact of the Gamma variant during its emergence, revealing distinct patterns in hospitalization rates, severity of illness, and outcomes. While the study sounds good, the presentation and the interpretation of some results can be misleading. In this reviewer opinion, the authors need to present/explain in a better/more detailed way the relationships between transmissibility and virulence (in terms of severity) associated to new variants. Some references to the molecular mechanism of infection should be added at least in the introduction to create more context for the presented study.

MAJOR CONCERNS

INTRODUCTION

While the authors present the world situation about virus spread they declare that D614G was the first reported mutation….associated with … increased transmissibility, severity, immune escape, and reduced vaccine efficacy.

In this reviewer opinion the authors should briefly explain which are the molecular changes related to D614G mutation. In this regard the authors should briefly say something more about molecular mechanism of infection and interactors for providing more context to what they are declaring.

Concerning the molecular mechanism of infection and interactors the authors can add some sentences by reading (and citing) the following papers:

EPMA J, interactors: https://link.springer.com/article/10.1007/s13167-021-00267-w

Cell Reports, interactors: https://www.sciencedirect.com/science/article/pii/S2211124720311645?via%3Dihub

Concerning the D614G mutation (and new flexibility properties it confers to the spike protein) the authors can add some sentences by reading (and citing) the following papers:

STTT, spike flexibility: https://www.nature.com/articles/s41392-020-00369-3

Science, spike flexibility: https://www.science.org/doi/10.1126/science.abd5223

Appl Microb and Biotech, spike flexibility: https://link.springer.com/article/10.1007/s00253-021-11676-2

In addition, in this reviewer opinion, when the appearance of new variants resulted in an increased transmissibility, at the same time the virulence (in terms of symptoms severity) was decreased. At the same time a greater transmissibility can result in heavier spread that can statistically result in an increased fatality rate…but the normalization of deaths for number of infected people should reflect a lower virulence in terms of symptoms severity… All these aspects can be better explained in a manuscript to be published on “Virus” to not create confusion. In this regard, the authors have to correct their sentence about the generic increased transmissibility and virulence.. They can read something more about these aspects in the following papers:

EPMA J, transmissibility, virulence: https://link.springer.com/article/10.1007/s13167-021-00267-w

Nonlin Dyn, transmissibility, virulence: https://link.springer.com/article/10.1007/s11071-021-06705-8

In the supplementary file folder, Supp. Fig. 2 is not reported. Were the authors referring to Supp. Tables? If not, please add the missing supp. Fig.

RESULTS/DISCUSSION sections

In relation to the observed slightly greater need of use ICU in case of infection with gamma variant, the authors should also discuss, considering their analysis and the above reported papers, the relationships between the higher severity associated with the gamma variant compared with its relatively “lower” spread in the world compared to other variants (like the Indian or the omicron).

The size of characters in figure 1 should be increased, because it cannot be read. Please, check journal style, suggestions..

Author Response

See response in the file attached.

Reviewer 2 Report

Comments and Suggestions for Authors

Authors assessed the impact of SARS-CoV-2 Gamma variant on ICU admissions and developed the PCR genotyping method to detect variants. This study includes important data to prepare for the appearance of new variants in the point of public health. Also, PCR genotyping to detect SARS-CoV-2 variants is a useful tool independent of equipment and cost.

Reviewer 3 Report

Comments and Suggestions for Authors

The present work by Vasques Nonaka and colleagues focused on epidemiological and clinical data related to SARS-CoV-2 Gamma variant circulation in Northeast Brazil, in the period December 2020 – March 2021: during this time-lapse, an increase in COVID-19 cases and related deaths were registered in Bahia state, as well as the Gamma variant emerged in Salvador. The authors reported the results obtained from their standard diagnostic tests, plus strain characterization by specific RT-PCR, and they coupled these data with clinical outcomes, especially for patients admitted in the Intensive Care Unit: a more severe clinical picture, higher inflammation and a decrease in subjects hospitalized age were associated to Gamma variant in this setting.

The work is quite good written, even a review should be made to fix some mistake (examples: poor syntax in lines 53-57; uniform usage of gender in place of sex; verb missing in lines 164-165; usage of n(%) in tables; “significant Notably” in line 290). In addition, the first references (1-6) are poorly related to the context, and the n°5 sounds like an inappropriate self-citation.

Here follow some more specific concerns on the paper.

- In lines 59-64, the main scope of genomic surveillance appears to be the geographical virus distribution: even it is true, the link to clinical outcomes and new viral characteristics (i.e.: immune escape, virulence, transmissibility) must be reported, having be a key feature of SARS-CoV-2 pandemic.

- The variants naming is confounding: one among Greek letter, Nextstrain clade or WHO/PANGO lineage method should be adopted.

- The paragraph on PCR genotyping (line 79-92) is obsolete. This approach was reliable when only few variants were present in the population, while the rapid virus evolution claims for sequencing approach, in order to promptly detect new mutations: the present epidemiological situation would not benefit of a RT-PCR based approach for variants detection.

- The study was conducted in a population of subjects tested between December 2020 and March 2021, approximately 3 years ago; in the meanwhile, Gamma variant disappeared, new ones had arisen and gone, and thousands (perhaps millions) of epidemiological and clinical data on variants were collected, analysed and published. The scientific soundness of the present work is really poor. Moreover, the authors previously published a work on quite the same cohort (DOI:10.1016/j.ijid.2021.08.003 1201-9712), in absence of RT-PCR and non virological laboratory data: however, the new information lack in novelty, with limited adds to the previous article.

- From the methods point-of-view, the choice of primers and probe is not well explained: is the RT-PCR specific for Gamma? Why other variants were not included? Why two different kits were used? How the samples for sequencing were selected?

- How the laboratory parameters were selected? It looks like that the criterium “the more is better” was followed. In COVID-19 is well known that inflammation, cytokines storm and coagulation are key factors in the clinical evolution: for example, interleukin 6 was associated to more severe syndromes, especially in ICU setting; on the same way, hepatic markers are usually elevated in subjects with pre-existing liver dysfunction, while here only slightly increases were found. No discussion were reported on these aspects, that is critical to understand the importance of presented data, also considering that authors reported several statistically significant differences, without any reference to normal values.

- The patients were stratified in Gamma vs non-Gamma, but for a proper data interpretation, especially on significant values, other variants should be included.

Comments on the Quality of English Language

Quite fine, review sentences construction.

Round 2

Reviewer 2 Report

Comments and Suggestions for Authors

Authors have revised well.

Author Response

We thank the referee for their contributions.

Reviewer 3 Report

Comments and Suggestions for Authors

Thank you to authors for the clarifications and the improvements.

Still, some concern remain about novelty and results, that will be addressed to the editor for final evaluation.

Author Response

We thank the reviewers for their contributions.